# Improvement of Electrical Properties of Silver Nanowires Transparent Conductive by Metal Oxide Nanoparticles Modification

Wei Liu, Yuehui Hu *, Yichuan Chen , Zhiqiang Hu, Ke Zhou, Zhijian Min, Huiwen Liu, Lilin Zhan and Yinjie Dai

School of Mechanical and Electronic Engineering, Jingdezhen University of Ceramics Institute, Jingdezhen 333403, China
* Correspondence: 8489023@163.com; Tel.: +86-13979803899

**Abstract:** At present, silver nanowire transparent conductive films (AgNWs-TCFs) still have problems such as high resistance of AgNWs network nodes, uneven distribution of resistance and poor electrical performance stability, which restrict their commercial application. Different from chemical modification, in this paper, a method of modifying AgNWs-TCFs with metal oxide nanoparticles (MONPs) is proposed, that is, ZnO, SnO$_2$, Al$_2$O$_3$ and TiO$_2$ etc., four transparent metal oxides are used as targets respectively in a magnetron sputtering process, modifying the silver nanowire network wire–wire junctions and silver nanowire in AgNWs-TCFs using active MONPs generated by magnetron sputtering. A series of AgNWs@MONPs for the AgNWs@ZnO-TCFs, AgNWs@SnO$_2$-TCFs, AgNWs@Al$_2$O$_3$-TCFs and AgNWs@TiO$_2$-TCFs were obtained. A significant decrease in the resistance of AgNWs-TCFs through the modification of MONPs was shown. Respectively, the reduction of resistance was 75.6%, 70.4%, 53.2% and 59.8% for AgNWs@ZnO-TCFs, AgNWs@SnO$_2$-TCFs, AgNWs@Al$_2$O$_3$-TCFs and AgNWs@TiO$_2$-TCFs. Correspondingly, its non-uniformity of resistance distribution was 12.5% (18.2% before), 10.0% (17.1% before), 10.1% (24.3% before) and 10.6% (13.4% before), respectively, which markedly improved the uniformity of electrical property. Respectively, their failure voltages reach 16, 16, 15 and 16 (V), so accordingly, the electrical stability is considerably enhanced. In addition, the uniformity of temperature distribution was also significantly optimized with its temperature non-uniformity of 10.4%, 8.7%, 10.3% and 9.6%, respectively. Contrast that with AgNWs@MONPs, and the failure voltages and temperature non-uniformity of AgNWs-TCFs are 12 V and 40.6%.

**Keywords:** silver nanowire transparent conductive film; magnetron sputtering modification; metal oxide nanoparticles; electrical properties; thermal property

## 1. Introduction

As a result of the high optical transmittance (>90%) of silver nanowires (AgNWs), transparent conductive films (AgNWs-TCFs) and their superior electro-conductivity and flexibility [1] provide an ideal and flexible alternative to overcome the brittleness of In2O3:Sn transparent conductive films (TCFs) [2–5]. However, due to the disordered and uneven distribution of the AgNWs coated on the transparent substrate, and the complexity of the mass and heat transfer processes during the annealing process, the AgNWs distribution will exhibit a "coffee ring" phenomenon [6,7], which makes the sheet resistance of AgNWs-TCFs nonuniform. In addition, the wire–wire junctions in the AgNWs-TCFs network are not tightly contacted, resulting in high junction resistance and uneven electrical properties [8–11]. Therefore, the electromigration and Joule heating will be unevenly distributed, and it is easy to cause local silver nanowires to fuse, that is, the electrothermal failure of the conductive channels of the silver nanowire network, causing the failure of AgNWs-TCF [12]. At the same time, due to the large specific surface of silver nanowires, the water–oxygen reaction and sulfidation reaction are prone to occur in the air, which

seriously degrades the electrical conductivity [13]. These electrical stability and load reliability problems of AgNWs-TCFs have become the bottlenecks restricting the application of AgNWs-TCFs to flexible electronic devices, which have become a hot research topic for scientists [14–23].

Aiming at the problem of high junction resistance of AgNWs-TCFs, Kin Wai Cheuk et al. [24] and Gao Jinwei et al. [25] reported the research results of junction-free AgNWs-TCFs, respectively. Kin Wai Cheuk et al. fabricated junction-free AgNWs-TCFs using a thermal evaporation technique in the fractured regions of polymer templates with conformal coatings, which can withstand 2.6 times higher bias current compared to ordinary AgNWs-TCFs. Gao Jinwei et al. developed two different junction-free highly conductive and transparent micro scaffold networks using biomimetic technology, with special fractal properties of low-voltage systems. However, the junction-free AgNWs-TCFs developed by Kin Wai Cheuk and Gao Jinwei et al. still face challenges in terms of high transmittance. Hao Chen et al. [26] reported that gold nanoparticles (AuNPs) modified silver nanowire networks wire–wire junctions by the dip-spraying method, and the obtained AgNW@AuNP thin films showed excellent thermal stability (350 °C). In addition, Its sheet resistance is 8.1 $\Omega$/sq, and its absorbance is less than 6%. Similarly, Bharat Sharma et al. [27] used Ag-NWs and CoNPs mixed solution spin coating method to modify silver nanowire networks wire–wire junctions with CoNPs to obtain AgNWs-CoNPs composite thin films with the best transmittance of 94% and the sheet resistance of 52 $\Omega$/sq. Hu Yuehui et al. reported [28] that the AgNWs-TCFs were obtained with a sheet resistance of 20 $\Omega$/sq and uniform electrical properties by using the water mist method to make the silver nanowire networks wire–wire junctions in close contact through capillary force. However, the above-mentioned method of making silver nanowire networks wire–wire junctions in close contact still faces challenges in solving the degradation of electrical properties caused by the water–oxygen reaction of AgNWs-TCF in air.

Different from the modification of AgNWs-TCFs in the above-mentioned literatures [26,27], this paper proposes a method to modify the wire–wire junction of AgNWs-TCF with active metal oxide nanoparticles (MONPs) generated by controlling the magnetron sputtering process (especially sputtering time and power), the MONPs sputtered from the target stop sputtering in its nucleation stage, realizing the modification of AgNWs-TCFs. Four transparent metal oxides, $TiO_2$, $SnO_2$, $Al_2O_3$ and ZnO, were used as targets respectively. It is expected that active MONPs will modify the wire–wire junction of AgNWs-TCFs, which not only reduces the junction resistance of AgNWs-TCF, but also forms a metal oxide layer on the surface of silver nanowires to wrap the silver nanowires, thereby improving the oxidation resistance of AgNWs-TCF.

## 2. Materials and Methods

The ceramic substrate cleaned by the semiconductor cleaning process was placed on a spin coater (Institute of Electrics, Chinese Academy of Sciences, Beijing, China). The thin film preparation process was performed according to the technical route shown in Figure 1: the AgNWs solution was diluted by mixing 1 mL of silver nanowires and 3 mL of isopropanol. The diluted AgNWs solution was extracted with a plastic-tip dropper and dropped on the substrate. After rotating at 600 rpm for 8 s first, the AgNWs wet film was obtained by rotating at 2000 rpm for 30 s. The AgNWs wet film was placed on a 100 °C heating-table for heat treatment for 5 min. The above process was repeated three times to obtain ceramic substrate AgNWs-TCFs with sheet resistances of 24.48 $\Omega$/sq, 41.28 $\Omega$/sq, 39.12 $\Omega$/sq and 53.28 $\Omega$/sq, respectively, marked as #1, #2, #3 and #4 sample. $TiO_2$, $SnO_2$, $Al_2O_3$, ZnO metal oxide nanoparticles, produced by magnetron sputtering method, were sputtered on the surfaces of #1, #2, #3 and #4 samples respectively to wrap the silver nanowires and modify the AgNWs networks wire–wire junctions obtaining AgNWs@$TiO_2$-TCFs, AgNWs@$SnO_2$-TCFs, AgNWs@$Al_2O_3$-TCFs, AgNWs@ZnO-TCFs showing in Figure 1d. The magnetron sputtering process was set as follows: working pressure of 1.2 Pa, power of 120 W, argon flow of 40 sccm and sputtering time of 15 min.

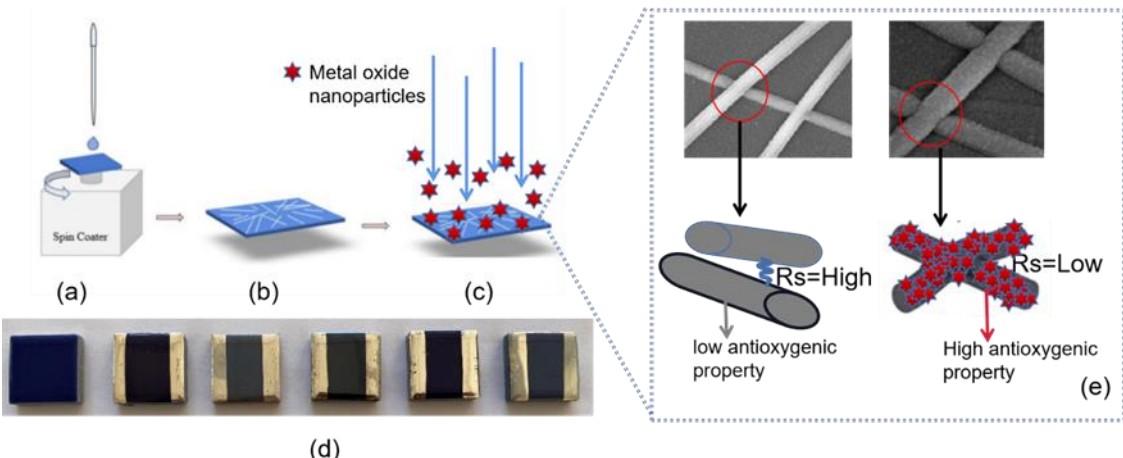

**Figure 1.** Schematics illustrating the MONPs modify of AgNWs network by magnetron sputtering method and its positive impact on AgNW-TCF. (**a**) The fabrication of AgNWs-TCFs by spin coating. (**b**) AgNWs-TCFs. (**c**) The MONPs modify the AgNW network by magnetron sputtering method. (**d**) The ceramic substrate and AgNWs@ MONPs-TCFs samples, from left to right, are ceramic substrate, AgNWs-TCF, AgNWs@SnO$_2$-TCF, AgNWs@Al$_2$O$_3$-TCF, AgNWs@TiO$_2$-TCF and AgNWs@ZnO-TCF, respectively. (**e**) Carrier transport across the wire−wire junction and the layer wrap on AgNW in AgNWs-TCFs before (left) and after (right) modified by MONPs.

The surface morphology of the films was analyzed by SU-8010 scanning electron microscope (HITACHI, Tokyo, Japan) and the KDY-1 resistivity/resistance four-probe was used to test the sheet resistance of the film. In addition, the withstand voltage of samples was tested using PRECI-200 series bench table source meter (PRECISE INSTRUMENT, Wuhan, China), and the temperature was measured using a Guide-P120V thermal imager (GUIDE, Wuhan, China). All operations and measurements were carried out at room temperature.

## 3. Results and Discussion

### 3.1. Electrical Properties of AgNWs Transparent Conductive Films

Figure 2 shows the test results of the sheet resistance of the #1~#4 samples. It can be seen that the resistance of the AgNWs-TCFs modified by MONPs is significantly reduced. Among them, the resistance reduction ratio of AgNWs@ZnO-TCFs and AgNWs@SnO$_2$-TCFs is larger—75.6% and 70.4%, respectively—while the resistance reduction ratio of AgNWs@Al$_2$O$_3$-TCFs and AgNWs@TiO$_2$-TCFs is smaller at 53.2% and 59.8%, respectively.

According to Yonggao Jia et al. [29], the non-uniformity is defined with the relative standard deviation as:

$$NUF = \sqrt{\frac{1}{n}\frac{\sum_{i=1}^{n}\left(R_i - \overline{R}\right)^2}{\overline{R}^2}} \qquad (1)$$

where $n$ is the number of measurements on the film of different sites, and $R_i$ and $\overline{R}$ are the measured resistance and the average resistance of all the measurements, respectively. The obtained distribution of resistance's relative standard deviation of AgNWs@ZnO-TCFs, AgNWs@SnO$_2$-TCFs, AgNWs@Al$_2$O$_3$-TCFs and AgNWs@TiO$_2$-TCFs were 12.5% (18.2% before modification), 10.0% (17.1% before modification), 10.1% (24.3% before modification) and 10.6% (13.4% before modification), respectively, indicating that the resistance's relative standard deviation of AgNWs-TCFs decreased significantly after MONPs modification. By comparing the mapping images before and after MONPs modification treatment of samples #1 to #4 showing in Figure 3, it is further confirmed that the resistance's relative standard deviation of AgNWs-TCFs after MONPs modification treatment is obviously reduced.

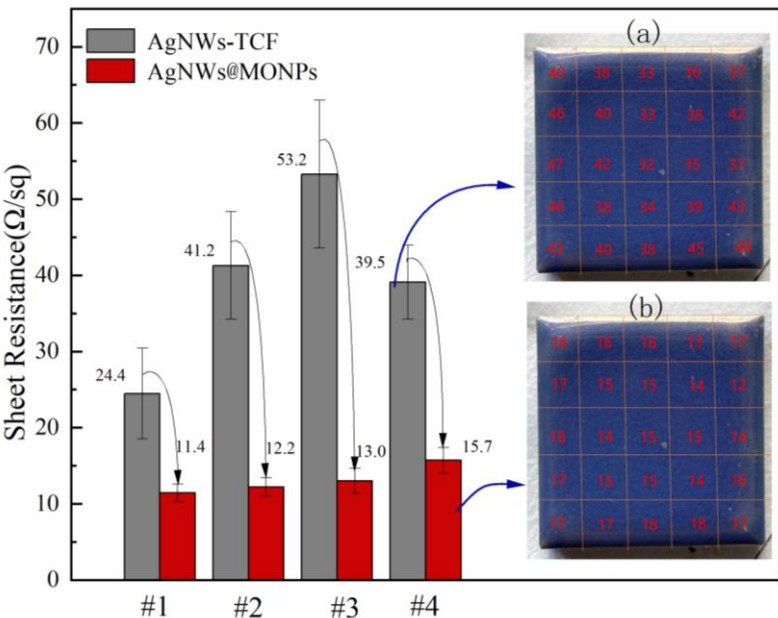

The marks of #1, #2, #3 and #4 represent AgNWs-TCF before and after being modified with Al$_2$O$_3$, SnO$_2$, ZnO and TiO$_2$ nanoparticles, respectively.

**Figure 2.** The sheet resistance of AgNWs-TCFs before and after MONPs modification. The inset (**a**) is the physical diagram of bare AgNWs-TCFs resistance test location, and inset (**b**) is the physical diagram of AgNWs @TiO$_2$-TCFs resistance test location.

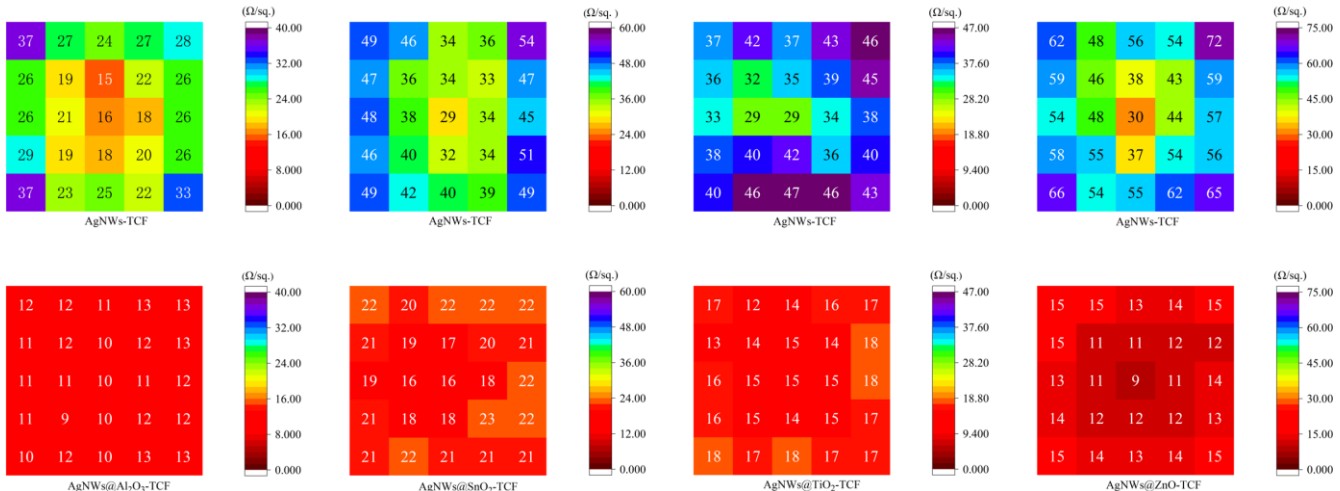

**Figure 3.** Mapping of samples #1~#4 before and after MONPs modification.

The resistance of AgNWs-TCFs is mainly derived from the junction resistance of the wire–wire junction in the silver nanowire network. AgNWs-TCFs prepared by the spin coating method have poor wire-to-wire junction contact in the silver nanowire network, resulting in higher junction resistance. We believe that the active MONPs sputtered from the metal oxide target by magnetron sputtering resulted in good wire-to-wire junction contact after modifying the AgNWs-TCFs, thereby reducing its resistance. Figure 4 shows the SEM images and cross-sectional views of the silver nanowire network before and after MONPs modification sputtered from targets Al$_2$O$_3$, SnO$_2$, TiO$_2$, ZnO, respectively. It can be seen from the figure that there is no obvious metal oxide film formed on the surface of AgNWs-TCFs, but it can be seen from the cross-sectional view that MONPs have obvious effects on the modification of AgNWs and wire–wire junctions in AgNWs-TCFs. At the same time, it can be seen that under the same sputtering parameters, the modification effects of active

MONPs sputtered from different metal oxide targets are quite different. Figure 4c,e are SEM images after modification with $SnO_2$ and ZnO nanoparticles, corresponding to their thicknesses of wrapping-layer of 43 nm and 33 nm respectively, which indicate that their diameters are significantly increased, and their wire–wire junctions are in close contact, suggesting the $SnO_2$ and ZnO nanoparticles have a significant effect on the modification of AgNWs-TCFs. Figure 4b,d are the SEM images of AgNWs-TCFs modified by MONPs sputtered from $Al_2O_3$ and $TiO_2$ targets, and the effect is not as good as $SnO_2$ and ZnO nanoparticles corresponding to their thickness of wrapping-layer of 14 nm and 7 nm respectively, but it can be seen from the cross-section SEM images inset in Figure 4b,d that the wire–wire junction of silver nanowires are in close contact after being modified by $Al_2O_3$ and $TiO_2$ nanoparticles, and the surface of the silver nanowires is wrapped well.

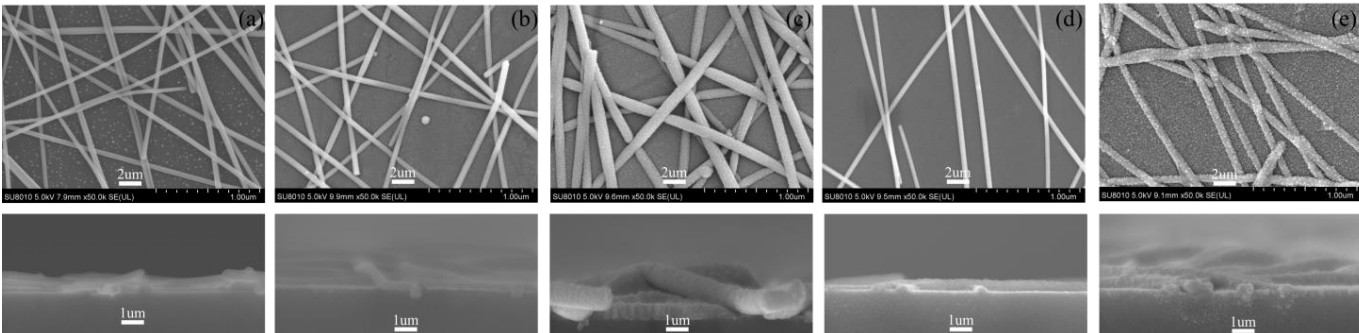

**Figure 4.** SEM image of the silver nanowire network of samples. (**a**–**e**) are the SEM images of AgNWs-TCFs, AgNWs@$Al_2O_3$-TCFs, AgNWs@$SnO_2$-TCFs, AgNWs@$TiO_2$-TCFs and AgNWs@ZnO-TCFs, respectively. The second row shows SEM images of cross-sections.

The reason for the different modification and encapsulation effects of different MONPs on AgNWs-TCFs may be related to the adsorption and migration of these MONPs on the silver nanowire network and ceramic substrate. The adsorption of MONPs on the substrate depends on the activated-collision for the substrate and the MONPs by sputtering gas (Ar), while the migration ability of the MONPs depends on the excitation energy required for the MONPs to migrate from the A to the adjacent B position. Under the condition of certain sputtering parameters, the adsorption capacity of different MONPs on the substrate is not different, but the excitation energy required for the MONPs with large atomic number to migrate from the A to the adjacent B position is larger, and it is not easy to be migrated to the adjacent position. In sequence, the thickness of the amassing-layer with large atomic number MONPs is larger than that of a small one, resulting in better modification and encapsulation effect on AgNWs-TCFs than that of the small one.

In summary, although the modification effect of active MONPs sputtered from different metal oxide targets is quite different under the same sputtering parameters, the highly active MONPs produced by the magnetron sputtering method can not only well realize the modification of silver nanowires networks which results in good wire–wire junction contact closely, reducing AgNWs-TCFs resistance, but also can be well wrapped on the surface of the silver nanowires. This encapsulation on the surface of silver nanowires with MONPs will play an important role in improving the anti-oxidation performance and bias current stability of AgNWs-TCFs.

### 3.2. Electrothermal Properties of AgNWs Transparent Conductive Films

The voltage which was applied across the electrodes for each sample was gradually increased in steps of 1V (the voltage was increased after reaching thermostabilization), until the conductivity of the sample failed, namely the sample was non-conductive. The relationship between temperature and voltage is shown in Figure 5. It can be seen that the temperature of AgNWs-TCFs, AgNWs@$SnO_2$-TCFs, AgNWs@$TiO_2$-TCFs, AgNWs@ZnO-TCFs and AgNWs@$Al_2O_3$-TCFs increases slowly with increasing voltage at voltages below 4 V,

while their temperature increases rapidly with the increase of voltage until the conductive channel of the sample fails as the voltage is greater than 4 V, corresponding failure voltages are 12, 16, 16, 16 and 15 V respectively. It indicated that AgNWs wrapped by MONPs can effectively improve the bias current stability owing to the effect of good wrapping on the surface of silver nanowires.

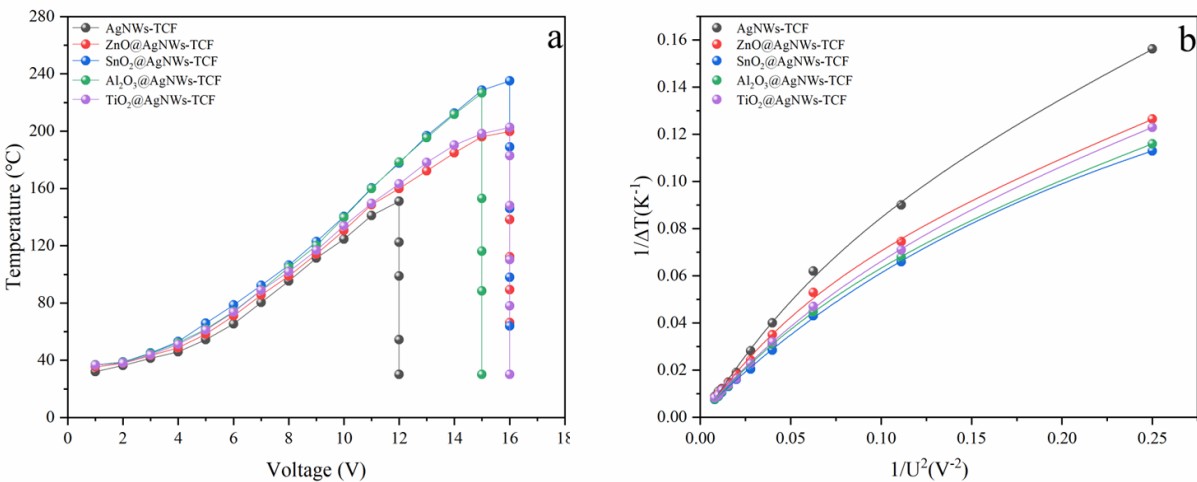

**Figure 5.** (**a**) Response of temperature with load voltage. (**b**) corresponds to the relationship between $1/\Delta T$ and $1/U^2$.

Figure 6 shows the relationship between power and radiation temperature. It can be seen from the figure that the heating efficiency of AgNWs-TCFs modified by MONPs is significantly improved. This may be related to the reduced heat transfer coefficient and temperature coefficient of resistance of AgNWs network after modification. According to the theoretical Formula (2) [30]:

$$T_s = T_0 + \frac{2P}{hA + \sqrt{4a^2p^2 + h^2A^2}} \tag{2}$$

where $T_s$ is the steady-state temperature, $T_0$ is the initial temperature, $\alpha$ is the temperature coefficient of resistance, $h$ is the total heat transfer coefficient including the AgNWs networks and substrate and $P$ is the input power. According to Figure 5b, there is little difference in the temperature coefficient of resistance at more than 5V voltages, and $T_s$ is mainly determined by the thermal conductivity. After being modified by MONPs, the silver nanowires or wire–wire junctions in the AgNWs network were wrapped or modified by metal oxides which reduced the heat transfer coefficient, resulting in the steady-state temperature $T_s$ improvement.

Figure 7 shows the thermal stability results of AgNWs-TCFs before and after modification with MONPs. It can be seen that in the unmodified AgNWs-TCFs, at a temperature of 92 °C for 7 h, the temperature drops significantly, indicating that most of the conductive channels of the silver nanowires have been fused or degraded as shown in the inset of Figure 7. After modification of AgNWs-TCFs, the temperature remained basically unchanged for 24 h at a temperature greater than 95 °C, and its thermal stability was significantly improved. This is because in the AgNWs-TCFs modified by MONPs, the silver nanowires or wire–wire junctions in the silver nanowire network are wrapped by metal oxides nanoparticles as shown in the cross-sectional SEM image in the second row of Figure 4, which plays a role in blocking the thermal oxygen reaction, thereby improving the thermal stability.

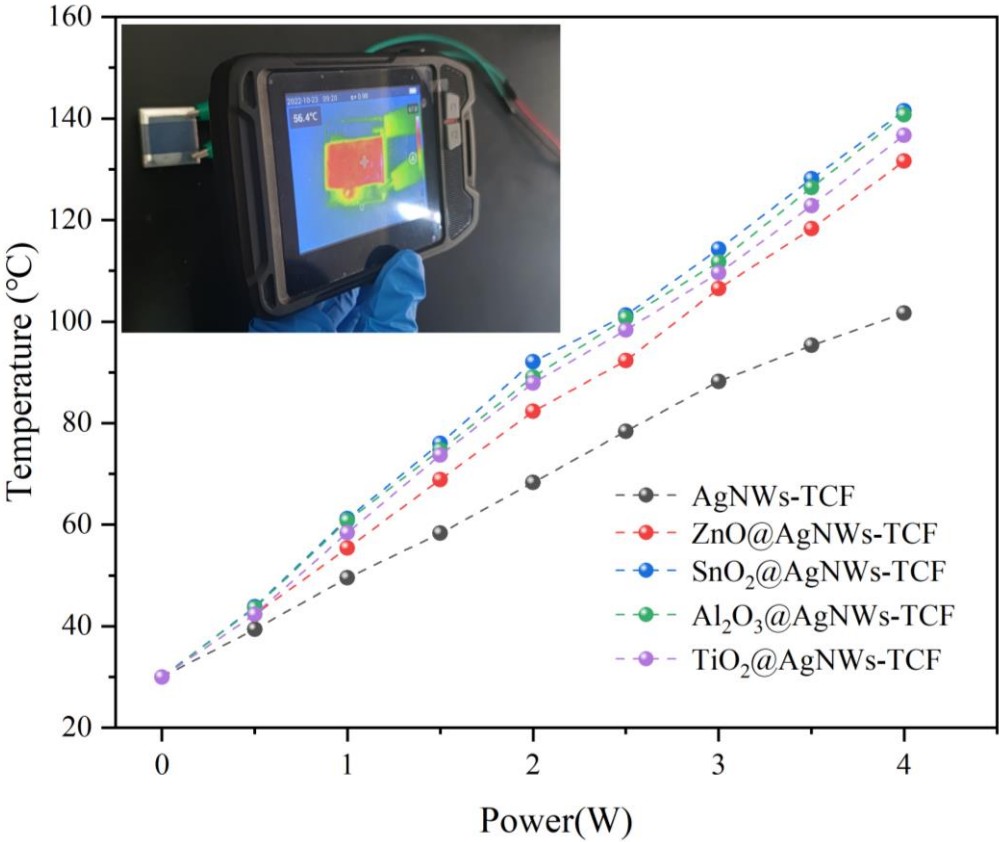

**Figure 6.** Relationship between electric power and temperature.

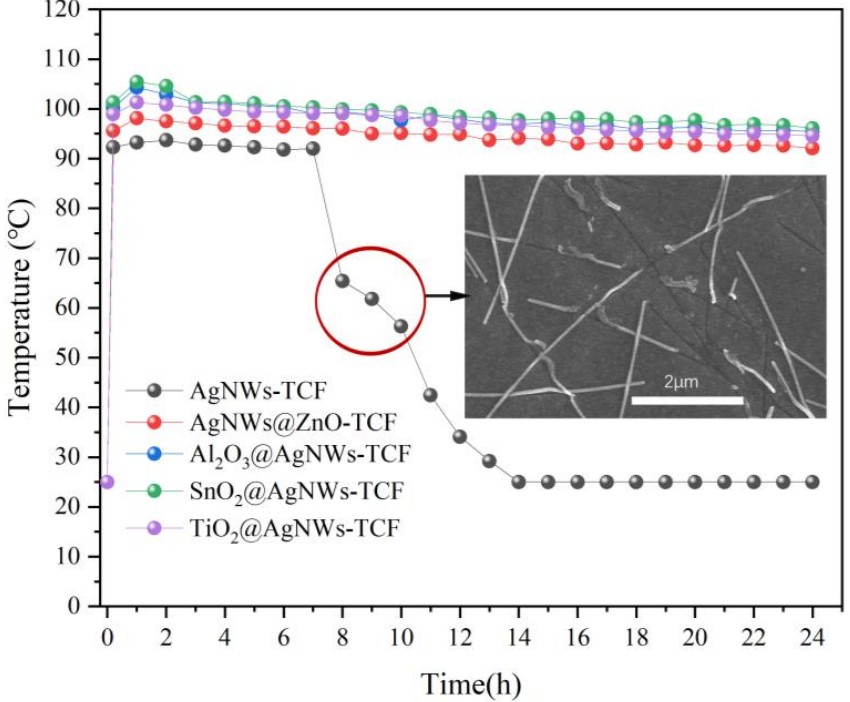

**Figure 7.** Thermal stability of AgNWs-TCFs before and after modification with MONPs. The inset is the SEM image of AgNWs-TCFs after its electrothermal failure.

Figure 8 is a three-dimensional model of the surface temperature distribution of AgNWs-TCFs. The model is based on the surface temperature distribution pictures of AgNWs-TCFs measured by a thermal imager and analyzed with Guide-ThermoTools. From Figure 8a, it can be seen that the surface temperature distribution of bare AgNWs-TCFs in the 3D model diagram is not uniform, while from Figure 8b–e, the surface temperature distribution of the AgNWs-TCFs modified by MONPs is becomes even more obvious. According to the theoretical formula [31]:

$$T_{non-uniformity} = \left[ \frac{T_{Max} - T_{Min}}{2T_{ave}} \right] \times 100\% \qquad (3)$$

where $T_{non-uniformity\ (\%)}$ is the temperature non-uniformity, $T_{max}$, $T_{min}$ and $T_{ave}$ are the maximum, minimum and average temperatures, respectively. The surface temperature non-uniformity of AgNWs-TCFs, AgNWs@Al$_2$O$_3$-TCFs, AgNWs@SnO$_2$-TCFs, AgNWs@TiO$_2$-TCFs and AgNWs@ZnO-TCFs is estimated to obtain the value of 40.6%, 10.3%, 8.7%, 9.6% and 10.4%, respectively. We believe that the reason for the improved uniformity of temperature distribution in AgNWs@MONPs-TCFs is that the modified AgNWs-TCFs have reduced the junction resistance of AgNWs network, improved the uniformity of its resistance distribution, and reduced the migration resistance of heat at the wire–wire junctions.

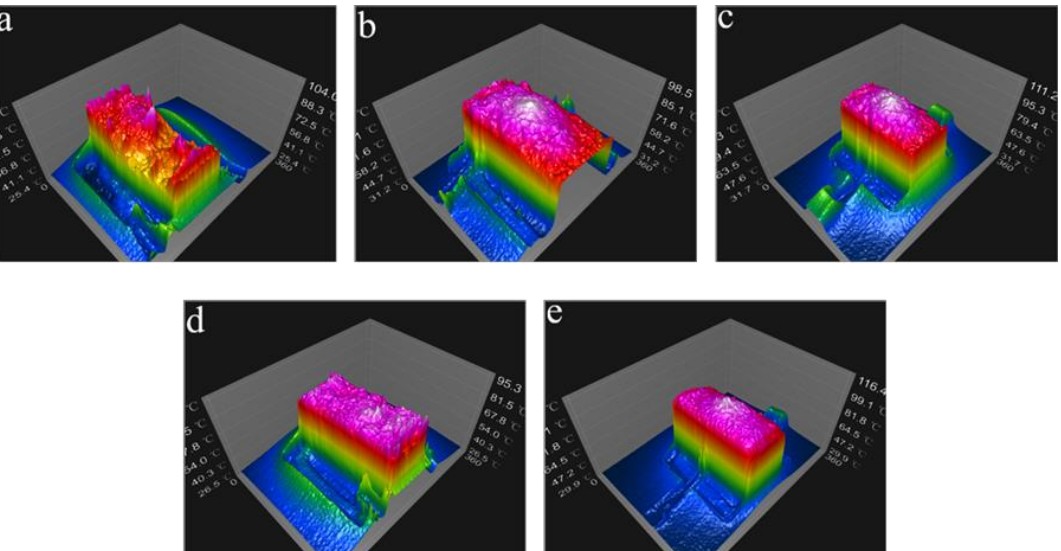

**Figure 8.** The 3D model of surface temperature distribution of AgNWs-TCFs. (**a**) The bare AgNWs-TCFs, (**b**) AgNWs@TiO$_2$-TCFs, (**c**) AgNWs@Al$_2$O$_3$-TCFs, (**d**) AgNWs@ZnO-TCFs, (**e**) AgNWs@SnO$_2$-TCFs.

## 4. Conclusions

By controlling the parameters of the magnetron sputtering process, the active MONPs sputtered from the metal oxide target can well realize the modification of the wire–wire junctions and the encapsulation of AgNWs at AgNWs networks. MONPs sputtered from four metal oxide targets including Al$_2$O$_3$, TiO$_2$, SnO$_2$, ZnO, etc., respectively, modified AgNWs-TCFs to obtain AgNWs@Al$_2$O$_3$-TCFs, AgNWs@TiO$_2$-TCFs, AgNWs @SnO$_2$-TCFs and AgNWs@ZnO-TCFs, which compared with bare AgNWs-TCFs, has a significantly lower resistance and significantly improved resistance distribution uniformity, electrical performance stability and electrothermal temperature distribution uniformity. The research results preliminarily solve the bottleneck problem of the silver nanowire-based transparent conductive film in the industrial application of flexible electronic devices and transparent heaters, which have important industrialization prospects.

**Author Contributions:** Conceptualization, Y.H. and W.L.; Formal analysis, Y.H.; Investigation, Y.C., Z.H. and K.Z.; Methodology, Y.H., W.L., Z.M., H.L. and Y.D; data curation, Y.H., W.L., K.Z. and Z.M.; Supervision, Y.H., W.L, L.Z. and K.Z.; Writing—original draft, Y.H. and W.L.; Writing—review and editing, Y.H., W.L. and Y.C. All authors have read and agreed to the published version of the manuscript.

**Funding:** This research was funded by the National Natural Science Foundation of China (Grant No. 62041405), the Natural Science Foundation of Jiangxi Province, China (Grant No. 20202BAB202011), the Key R & D Program of Jiangxi Province, China (Grant No. 20192BBE50056 and 20171BBE50053), the Education Bureau of Jiangxi Province, China (No. GJJ211328, GJJ211319).

**Institutional Review Board Statement:** Not applicable.

**Informed Consent Statement:** Not applicable.

**Data Availability Statement:** The data presented in this study are available on request from the corresponding author.

**Conflicts of Interest:** The authors declare no conflict of interest.

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
