# Peer review of "Improvement of Electrical Properties of Silver Nanowires Transparent Conductive by Metal Oxide Nanoparticles Modification"

_coatings, doi:10.3390/coatings12121816_

Round 1

Reviewer 1 Report

This paper describes a method of modifying AgNWs-TCFs with metal oxide nanoparticles (MONPs). It is an interesting research work that could be acceptable for publication after the following matters are addressed:

1.   Abstract

Line 16-22: very long and confusing sentence

Line 17: that: After MO” – change the capital letter

Line 23: confuse sentence

2.   Introduction

Line 29-33: very confuse sentence

Line 58: change the capital letter

3.   Materials and Methods

Why did you choose to obtain ceramic substrate AgNWs-TCFs with different sheet resistances before you covered it with the MONPs? In this way the starting ceramic substrate is different for all the MONPs.

4.   Results and discussion

The scale of figure 4 is too small and cannot be seen well.

You proved that modifying AgNWs-TCFs with MONPs improve the properties of the silver nanowires, but you don’t explain the different results between the four metal oxides used in the work. The papers seems a simple report of the results obtained without understanding them deeper. You should explore and explain better the different results between the metal oxides.

Reviewer 2 Report

The authors of this paper have presented a novel method to enhance electrical and thermal properties of silver nanowire transparent films, and the methods were verified experimentally. Design of the research, experimental methods, results, and discussion, are well explained. I have a few minor comments and questions to be considered by authors for improvements.

1) In line 192-200, temperature coefficient and thermal conductivity values are related with the steady state temperature, and it is explained with the inset in figure 5. I think it is better to improve the explanation by modifying the texts and adding another figure (take out the inset in figure 5) showing the temperature coefficient of the films.

2) In line 123, equation (1) shows the definition of non-uniformity. It is actually relative standard deviation (standard deviation/mean), so I think it is better to change line 122 like "the non-uniformity is defined with the relative standard deviation as: "

3) In figure 4, it is hard to see the cross section images because they are very small. Can you make them bigger?

4) There are some grammar errors and typos, so I highly recommend thorough proof reading to correct those errors. Some examples are:

line 84: By rotating at 600 rpm -> After rotating at 600 rpm

line 88: 1#, 2#, 3# -> #1, #2, #3

line 155: Al2O3 -> Al2O3 (subscripts)

Reviewer 3 Report

The authors studied the effect metal oxide nanoparticle coating on silver NWs. Metal oxide NPs were deposited by PVD and characterized using SEM. The modified AgNWs exhibit lower sheet resistance and improved resistance uniformity. They also exhibited better stability under applied voltage and thermal stress This manuscript, in content, can be accepted for publication in Coatings, only after appropriate revisions has been made according to the comments and questions of this referee given below.

1.        Provide the size and distribution of different metal nanoparticles. Are different electrical properties obtained with different metal nano particles coated AgNW due to their difference in size?

2.        In the materials and methods section, it is not clear how the Ag-NWs with different resistances are obtained. Explain in detail

3.        What is the rationale behind choosing AgNWs with different resistance for testing with different metal nanoparticles?  For example in Fig2, sample#1 with Al2O3, Sample#2 with SnO2 etc

4.        Cite other relevant works on metal oxide NPs coated Silver wire .  For example :
1. Nanoscale
, 2019,11, 19969-19979,

                  2. Sara Aghazadehchors. metallic nanowire networks : silver nanowire network stability                    enhancement using metal oxide coatings, Percolation onset of nano-object network. Chemical and Process Engineer[1]ing. Université Grenoble Alpes [2020-..]; Université de Liège, 2021. English. ffNNT : 2021GRALI044ff. fftel-03628360

Round 2

Reviewer 1 Report

I have no more comments to add.